# Characteristics and Evolution of Leaf Epidermis in the Genus *Amana* Honda (Liliaceae)

Xin Zeng [1,†], Meizhen Wang [1,†], Minqi Cai [2], Pengcheng Luo [3], Matthew C. Pace [4] and Pan Li [1,*]

1 Laboratory of Systematic & Evolutionary Botany and Biodiversity, College of Life Sciences, Zhejiang University, Hangzhou 310058, China; 3200101995@zju.edu.cn (X.Z.); 12207150@zju.edu.cn (M.W.)
2 Shanghai Science and Technology Museum, Shanghai 200127, China; caimq@sstm.org.cn
3 Wuxi Biologics (Cayman) Inc., Hangzhou 310018, China; luo_pengcheng@outlook.com
4 New York Botanical Garden, 2900 Southern Blvd., Bronx, New York, NY 10348, USA; mpace@nybg.org
* Correspondence: panli@zju.edu.cn
† These authors contributed equally to this work.

**Abstract:** *Amana*, commonly known as 'East Asian tulips', has recently been found to harbor cryptic diversity due to recent field work and systematic investigations. In this study, we included 64 populations from all 12 *Amana* species and performed microscopic observations of their epidermal morphology. The leaf epidermis stomatal distribution of *Amana* can be characterized into three types: dense stomata (>10/per view or 263/mm²), sparse stomata (<10/per view or 263/mm²), and stomata absent. The epidermal cells of *Amana* can be characterized into four types: rectangular, long rectangular, nearly rectangular, and rhombic. The anticlinal wall morphology of the epidermal cells can be characterized into three types: linear, wavy, and nearly linear with mixed shallow waves. All the results were helpful for classification of *Amana* species. According to the reconstruction of ancestral characters analyses, the common ancestor of *Amana* is most likely to have leaves with dense stomata on both sides, and epidermal cells that have linear vertical walls.

**Keywords:** character evolution; Liliaceae; morphology; stoma

## 1. Introduction

Even with the rapid development of new molecular tools to examine biodiversity, morphology remains one of the most important and decisive bases for species identification and classification. Identifying differences in the structure of the leaf epidermis has long been an important method for plant classification from the perspective of morphology. The differences in cell arrangement, stomatal density, and cell morphology of the leaf epidermis are strong indications of species identification. For example, Fan et al. [1] used epidermis cell morphology, cuticular wax, stomatal distribution, morphology, and the stomatal index to classify *Holcoglossum* Schltr. Jia et al. classified tree peonies based on stomatal morphology, stomatal density, vertical wall morphology, and trichome morphology [2].

The genus *Amana* Honda (Liliaceae) comprises about 12 perennial herbaceous species (ten already published, including three newly published: *A. nanyueensis* P. Li and L.X. Liu, *A. tianmuensis* P. Li and M.Z. Wang, and *A. hejiaqingii* M.Z. Wang and P. Li, and two new species in prep.: *A. polymorpha* ined. and *A. yunmengensis* ined.) that are restricted to East Asia [3–11]. The species of this genus are often called 'East Asian tulips', and once belonged to the genus *Tulipa* L. However, with the addition of more morphological (*Amana* having 2-3(-4) opposite or verticillate bracts and a longer style as long as the ovary while *Tulipa* does not), molecular, and geographical distribution evidence, these species were segregated as the distinct genus *Amana* [12–14]. Although the species of *Amana* have important medicinal and horticultural value, studies on its morphology are still lacking.

Previous studies showed significant differences in leaf epidermis characteristics among different species in the genus. For example, *Amana edulis* has linear anticlinal epidermal

cell walls, and both the adaxial and abaxial leaf epidermis have stomata. By contrast, *A. kuocangshanica* has typical wavy anticlinal epidermal cell walls and an adaxial epidermis lacking the stomata [15]. These findings make it possible to classify the *Amana* species with leaf epidermis characteristics, which may become an important basis for the classification and identification of *Amana* in the future.

In this study, we included 64 populations from all 12 *Amana* species and performed microscopic observations of their epidermal morphology. This study aims to provide support for subsequent morphological and systematic studies of *Amana*. In addition, it will deepen our understanding of the origin and evolution of the genus *Amana*.

## 2. Materials and Methods

### 2.1. Collection and Observation of Leaf Epidermis

We collected the fresh leaves of the relevant plants from the Botanical Garden of Zhejiang University (one leaf from 1–2 individuals per species), using tweezers to tear off the adaxial and abaxial epidermis of the leaves (about 2 × 2 mm each), and then made temporary slides in deionized water. We observed the epidermis from several pieces (at least three per leaf) across different parts of the leaf, so as to follow the principle of comparison. The epidermal tissues were observed with a compound microscope (4 × 10) and their characteristics were recorded, such as cell morphology, stomatal distribution, and the morphology of the anticlinal wall. The samples are not stained; the blue or red colors in the results are due to the original white balance adjustment of the computer. All the voucher specimens were deposited in the Herbarium of Zhejiang University (HZU) and experiment samples were of the transplanted populations in the Botanical Garden of Zhejiang University.

### 2.2. Reconstruction of Leaf Epidermis Features

Based on the results obtained from microscopic observation of the leaf epidermis, the epidermal characteristics of different species were summarized and recorded in Table 1. The morphology of leaf epidermal cells can be summarized as rectangular, long rectangular, nearly rectangular, and rhombic, and these characteristics can occur simultaneously within the same sample in different regions of the leaf. The morphology of the anticlinal wall of leaf epidermal cells can be put into three types: linear, wavy, and nearly linear (or mixed with mild waves in more linear types). The epidermal stomatal distribution can be classified as dense stomata, sparse stomata, or stomata absent. These traits were then analyzed one by one using the BBM method in RASP software to reconstruct ancestral characteristics [15]. The phylogenetic tree of *Amana* was constructed by maximum likelihood (ML) analysis implemented in IQ-TREE v2.2.0 [16] with 1000 bootstraps based on the nuclear genes (unpublished data). *Erythronium japonicum* (SRA number: SRR14576596) was used as the outgroup.

**Table 1.** The sample information.

| Species Name | Sample Number | Latitude | Longitude | Elevation/m | Locality |
|---|---|---|---|---|---|
| *A. anhuiensis* (X.S. Shen) Christenh. | LJK61 | 30.7235 | 116.4537 | 1711 | China, Anhui Province, Qianshan County, Mt. Tianzhu |
| | LJK62 | 30.7235 | 116.4537 | 1711 | China, Anhui Province, Qianshan County, Mt. Tianzhu |
| | LJK63 | 29.0967 | 115.5767 | 703 | China, Jiangxi Province, Yongxiu County, Mt. Yunju |
| | WMZ1499 | 29.0968 | 115.5768 | 711 | China, Jiangxi Province, Yongxiu County, Mt. Yunju |
| *A. baohuaensis* B.X. Han, Long Wang and G.Y. Lu, | LJK31 | 31.7883 | 119.2961 | 107 | China, Jiangsu Province, Jurong City, Mt. Mao |
| | LP207885 | 32.1385 | 119.2762 | 259 | China, Jiangsu Province, Zhenjiang City, Mt. Chaohuang |
| | LP207895 | 32.0559 | 118.5476 | 346 | China, Jiangsu Province, Nanjing City, Doushuai Temple |
| | LP207904 | 31.2614 | 119.7528 | 82 | China, Jiangsu Province, Yixing City, Songshan Village |
| | WMZ1060 | 31.4657 | 117.7861 | 76 | China, Anhui Province, Wuwei City, Yanqiao Town |
| | WMZ1417 | 32.1007 | 118.5875 | 114 | China, Jiangsu Province, Nanjing City, Mt. Lao |
| | WMZ1423 | 31.6714 | 118.0888 | 74 | China, Anhui Province, Hanshan County, Chaibulao Village |

**Table 1.** *Cont.*

| Species Name | Sample Number | Latitude | Longitude | Elevation/m | Locality |
|---|---|---|---|---|---|
| A. edulis (Miq.) Honda | LJK12 | 29.4536 | 120.2869 | 331 | China, Zhejiang Province, Zhuji City, Banqiu Village |
| | LJK38 | 31.2146 | 119.6959 | 247 | China, Jiangsu Province, Zhangzhu County, Mt. Longchi |
| | LJK54 | 31.0870 | 119.3347 | 308 | China, Anhui Province, Guangde City, Xiaojiawan |
| | LJK9 | 30.3864 | 118.2289 | 97 | China, Zhejiang Province, Changxing County, Biyan Temple |
| | LP173055A | 30.9435 | 120.0380 | 305 | China, Henan Province, Nanyang City, Mt. Du |
| | LP173072 | 33.0600 | 112.5786 | 183 | China, Jiangsu Province, Lianyungang City, Mt. Yuntai |
| | LP207881 | 34.7128 | 119.4206 | 99 | China, Anhui Province, Chuzhou City, Mt. Langya |
| | LP207884 | 32.2558 | 118.2789 | 177 | China, Henan Province, Tongbai City, Huaiyuan Town |
| | LP207887 | 32.4292 | 113.3197 | 269 | China, Jiangsu Province, Zhenjiang City, Mt. Mao |
| | LP207891 | 31.8154 | 119.3090 | 458 | China, Zhejiang Province, Ningbo City, Mt. Jin'e |
| | LP207905 | 29.6386 | 121.5596 | 247 | China, Jiangsu Province, Yixing City, Mt. Longchi |
| | LP207914 | 31.2146 | 119.6959 | 597 | China, Zhejiang Province, Jinhua City, Shuanglong |
| | WMZ1488 | 29.2092 | 119.6267 | 394 | China, Hubei Province, Guangshui City, Heilongtan |
| | WMZ1490 | 31.7980 | 114.0873 | 177 | China, Henan Province, Tongbai City, Huaiyuan Town |
| A. erythronioides (Baker) D.Y. Tan and D.Y. Hong | LJK24 | 29.7339 | 121.3428 | 169 | China, Zhejiang Province, Ningbo City, Fenghua District |
| | LJK26 | 29.7398 | 121.0869 | 845 | China, Zhejiang Province, Yuyao City, Mt. Siming |
| | LJK5 | 29.3735 | 121.5854 | 509 | China, Zhejiang Province, Ninghai County, Mt. Cha |
| A. hejiaqingii M.Z. Wang and P. Li, | LP173093 | 31.7207 | 115.5035 | 496 | China, Henan Province, Shangcheng County, Liluocheng Village |
| | LP207883 | 32.4036 | 113.3078 | 196 | China, Henan Province, Tongbai County, Chengjiao Village |
| | WMZ1487 | 31.7973 | 114.0872 | 529 | China, Hubei Province, Guangshui City, Heilongtan |
| | WMZ1489 | 32.3954 | 113.2989 | 248 | China, Henan Province, Tongbai County, Tayuan Temple |
| | WMZ1492 | 32.3066 | 113.4548 | 157 | China, Henan Province, Tongbai County, Mt. Bijia |
| | WMZ1714 | 31.9792 | 113.9154 | 185 | China, Henan Province, Xinyang City, Shihe District |
| | WMZ1716 | 31.8552 | 113.9298 | 199 | China, Hubei Province, Guangshui City, Santan |
| | WMZ1721 | 31.6234 | 115.4832 | 118 | China, Henan Province, Shangcheng County, Mt. Daniu |
| | ZXX19043 | 31.9792 | 113.9154 | 185 | China, Henan Province, Xinyang City, Shihe District |
| A. kuocangshanica D.Y. Tan and D.Y. Hong | LJK22 | 28.8150 | 120.9432 | 868 | China, Zhejiang Province, Linhai City, Mt. Kuocang |
| | LJK3 | 29.3752 | 121.5909 | 460 | China, Zhejiang Province, Ninghai County, Liyang Town |
| | LJK7 | 29.2582 | 121.3078 | 195 | China, Zhejiang Province, Ninghai County, Qiantong Town |
| A. kuocangshanica × A. latifolia | WMZ1448 | 28.5512 | 120.7998 | 413 | China, Zhejiang Province, Yongjia County, Mt. Sihai |
| A. latifolia (Makino) Honda | DSL01 | 27.9079 | 120.6970 | 324 | China, Zhejiang Province, Wenzhou City, Mt. Daluo |
| | WMZ1446 | 27.8273 | 120.3295 | 336 | China, Zhejiang Province, Rui'an City, Gaolou Town |
| A. nanyueensis P. Li and L.X. Liu | LP196220 | 27.2767 | 112.6746 | 1063 | China, Hunan Province, Hengyang City, Mt. Heng |
| | WMZ1464 | 27.2881 | 112.6932 | 1067 | China, Hunan Province, Hengyang City, Mt. Heng |
| A. polymorpha ined. | LJK10 | 29.5069 | 120.4369 | 438 | China, Zhejiang Province, Zhuji City, Liaozhai Village |
| | LJK11 | 29.4500 | 120.2860 | 413 | China, Zhejiang Province, Zhuji City, Banqiu Village |
| | LJK21 | 28.9799 | 120.5393 | 747 | China, Zhejiang Province, Pan'an County, Mt. Dapan |
| | LJK23 | 29.2525 | 121.0977 | 990 | China, Zhejiang Province, Tiantai County, Mt. Huading |
| | LP207908 | 29.3514 | 121.0259 | 629 | China, Zhejiang Province, Xinchang County, Xiaojiang Town |
| | WMZ1458 | 29.4500 | 120.2860 | 413 | China, Zhejiang Province, Zhuji City, Banqiu Village |
| A. tianmuensis P. Li and M.Z. Wang | LJK49 | 30.3884 | 118.2182 | 629 | China, Anhui Province, Huangshan City, Yuxiang Village |
| | WMZ1473 | 30.4728 | 117.8345 | 736 | China, Anhui Province, Qingyang County, Mt. Jiuhua |
| | WMZ1506 | 30.3884 | 118.2182 | 1620 | China, Anhui Province, Huangshan City, Mt. Huang |
| A. wanzhensis Lu Q. Huang, B.X. Han and K. Zhang | LJK39 | 31.2146 | 119.6959 | 272 | China, Jiangsu Province, Yixing City, Mt. Longchi |
| | LJK41 | 31.0864 | 119.3342 | 307 | China, Anhui Province, Guangde County, Xiasi Village |
| | LJK8 | 30.9452 | 120.0388 | 22 | China, Zhejiang Province, Changxing City, Biyan Temple |
| | WMZ1455 | 31.0871 | 119.3349 | 307 | China, Zhejiang Province, Zhuji City, Huangshan Town |
| | WMZ1460 | 29.4565 | 120.2906 | 572 | China, Zhejiang Province, Zhuji City, Huangshan Town |
| A. yunmengensis ined. | WMZ1707-1L | 29.5990 | 113.2556 | 161 | China, Hubei Province, Jianli County, Mt. Yanglin |
| | WMZ1707-2L | 29.5990 | 113.2556 | 161 | China, Hubei Province, Jianli County, Mt. Yanglin |
| | WMZ1709-1L | 29.6698 | 113.3250 | 8 | China, Hubei Province, Honghu County, Luoshan Town |
| | WMZ1709-3L | 29.6698 | 113.3250 | 8 | China, Hubei Province, Honghu County, Luoshan Town |

## 3. Results

### 3.1. Observations

3.1.1. Stomatal Distribution

Based on our microscopic observation, the leaf epidermis stomatal distribution in *Amana* can be characterized into three types: dense stomata (>10/per view or 263/mm$^2$),

sparse stomata (<10/per view or 263/mm$^2$), and stomata absent (Figure 1) (the area of each field of view is about 0.038 mm$^2$; for a more intuitive description, "per view" is used as the unit of area here). In the adaxial epidermis, stomata are lacking in *A. kuocangshanica*, and are absent or sparse in *A. polymorpha* ined. Individual samples of *A. erythronioides* appeared with both sparse and many stomata. All other species (*A. edulis*, *A. tianmunensis*, *A. nanyueensis*, *A. baohuaensis*, *A. wanzhensis*, *A. hejiaqingii*, *A. yunmengensis* ined., *A. anhuiensis*, *A. latifolia*) have many stomata in the adaxial epidermis (one sample from the *A. hejiaqingii* has a sparse stomatal distribution). In the abaxial epidermis, all species of *Amana* have dense stomatal distribution.

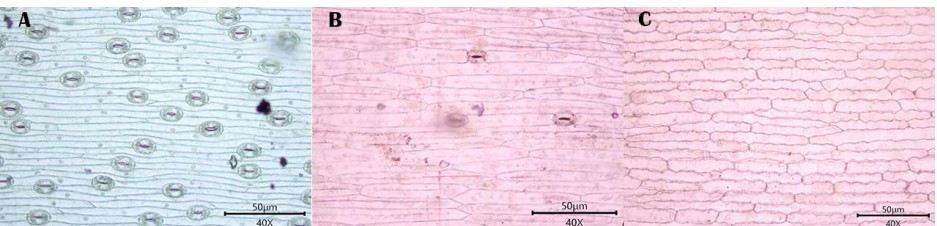

**Figure 1.** Graphic description of stomata: dense (**A**) from *A. nanyueensis*; sparse (**B**) form *A. hejiaqingii* and a absent (**C**) from *A. kuocangshanica*.

3.1.2. Morphology of Epidermal Cells

The epidermal cells of *Amana* can be roughly classified into four types: rectangular (the length-to-width ratio of cells is less than 5:1), long rectangular (the length-to-width ratio of cells is much greater than 5:1), nearly rectangular (cells appear as distorted or skewed rectangles), and rhombic (the cells are approximately geometrically diamond-shaped) (Figure 2). In the adaxial epidermis, these four cell forms are all present, but have different proportions in the same species, with the majority being rectangular cells (rectangular cells appear in at least one sample for all species). In the abaxial epidermis, these four types of epidermal cells have also been found, with different proportions in the same species, with long rectangles occupying the majority (as in the adaxial epidermis). In practical observation, the morphological changes of epidermal cells, especially the adaxial epidermis, are quite complex and sometimes cannot be easily described quantitatively.

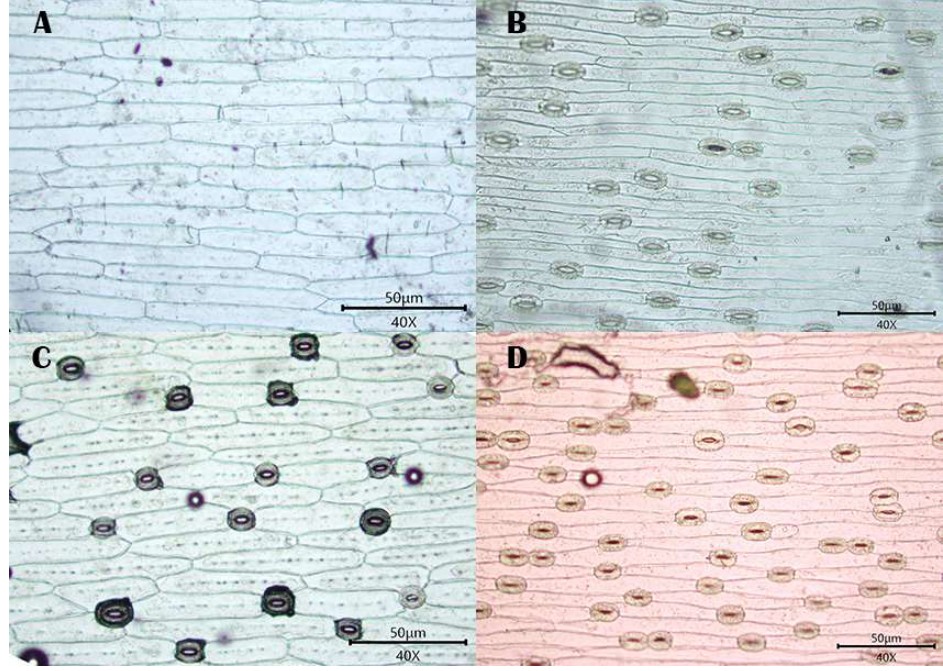

**Figure 2.** Graphic description of cells: rectangular (**A**) from *A. polymorpha* long rectangular (**B**) from *A. erythronioides*; nearly rectangular (**C**) form *A. hejiaqingii* and rhombic (**D**) from *A. latifolia* epidermal cells.

### 3.1.3. Morphology of the Anticlinal Wall of Epidermal Cells

The anticlinal wall morphology of the epidermal cells of *Amana* can be classified into three types: linear, wavy, and nearly linear with mixed shallow waves (Figure 3). In the adaxial epidermis we observed all three types of epidermal cell anticlinal walls. Wavy anticlinal walls were found in both *A. erythronioides* and *A. kuocangshanica*. In the samples of *A. nanyueensis*, all of the individuals were near linear. In *A. edulis*, *A. hejiaqingii*, and *A. polymorpha*, we observed both linear anticlinal circumferential walls and near linear anticlinal circumferential walls. All samples from other species (*A. tianmunensis*, *A. baohuaensis*, *A. wanzhensis*, *A. hejiaqingii*, *A. yunmengensis* ined., *A. anhuiensis*, *A. latifolia*) had straight anticlinal walls. In the abaxial epidermis, all samples had linear anticlinal walls (Figures 4–7, Table 2), except for one nearly linear anticlinal wall sample appearing in each of *A. edulis* (LP207891), *A. baohuaensis* (WMZ1060), and *A. erythronioides* (LJK26).

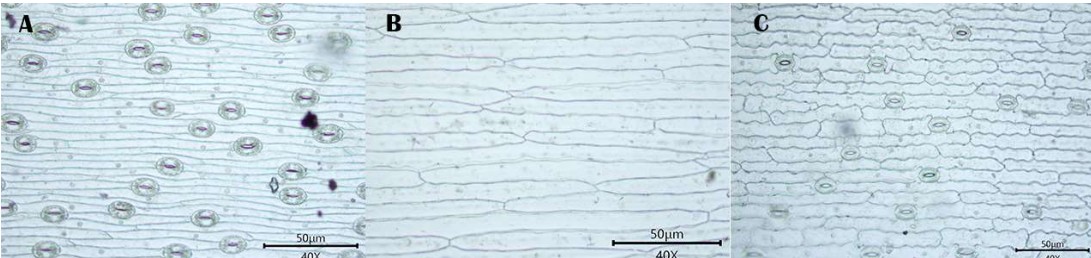

**Figure 3.** Graphic description of walls: linear (**A**) from *A. nanyueensis*; nearly linear with mixed shallow waves (**B**) form *A. polymorpha* and wavy (**C**) anticlinal from *A. erythronioides*.

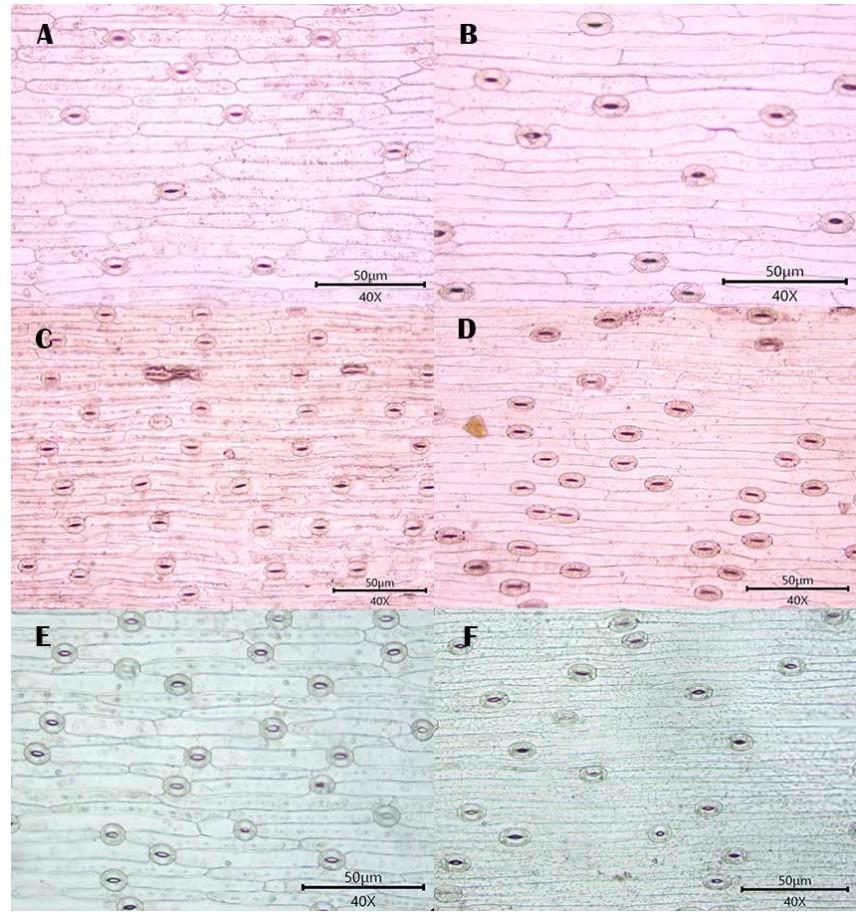

**Figure 4.** (**A**) Adaxial epidermis of *Amana anhuiensis*; (**B**) abaxial epidermis of *A. anhuiensis*; (**C**) adaxial epidermis of *A. baohuaensis*; (**D**) abaxial epidermis of *A. baohuaensis*; (**E**) adaxial epidermis of *A. edulis*; (**F**) abaxial epidermis of *A. edulis*.

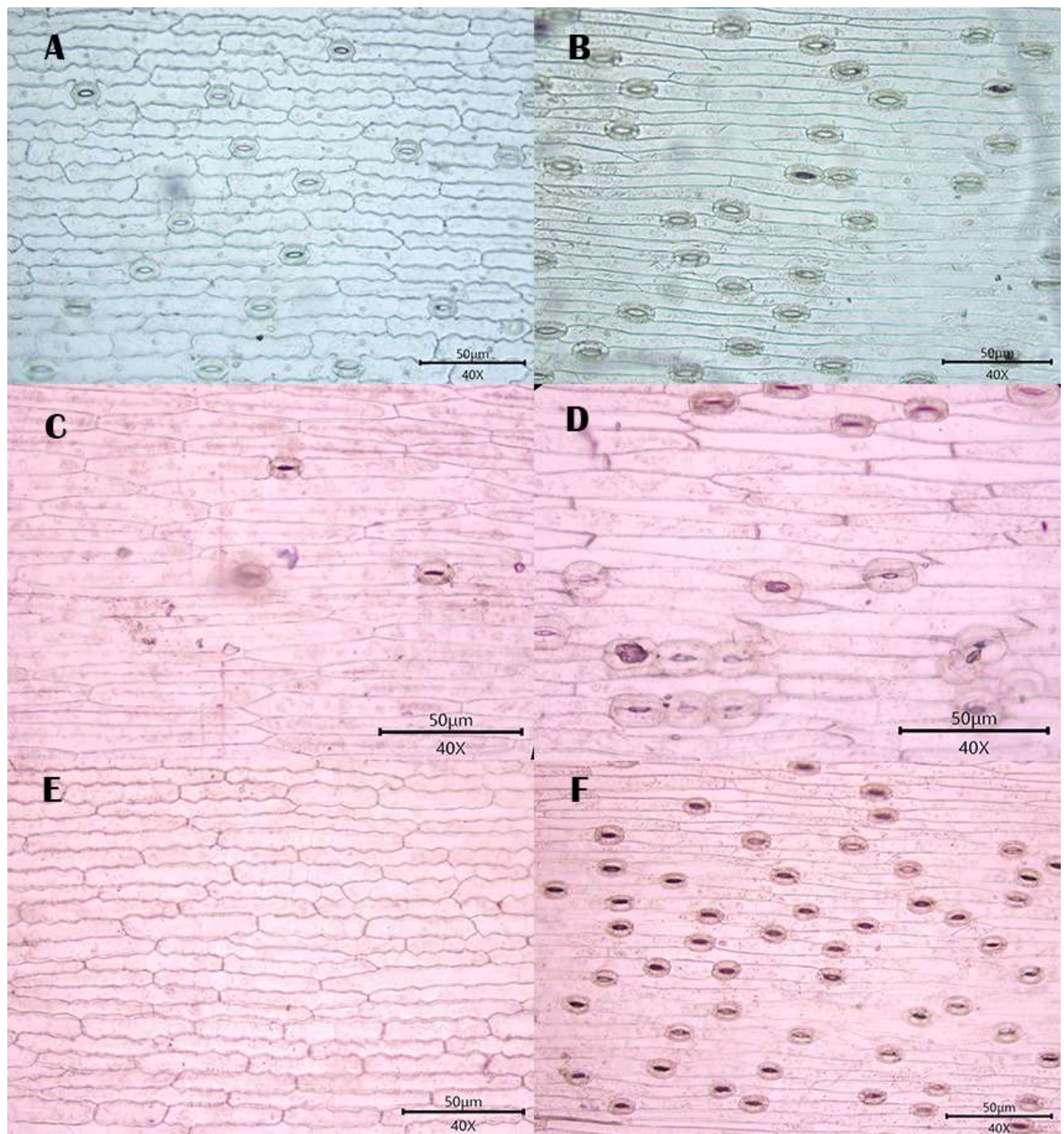

**Figure 5.** (**A**) Adaxial epidermis of *A. erythronioides*; (**B**) abaxial epidermis of *A. erythronioides*; (**C**) adaxial epidermis of *A. hejiaqingii*; (**D**) abaxial epidermis of *A. hejiaqingii*; (**E**) adaxial epidermis of *A. kuocangshanica*; (**F**) abaxial epidermis of *A. kuocangshanica*.

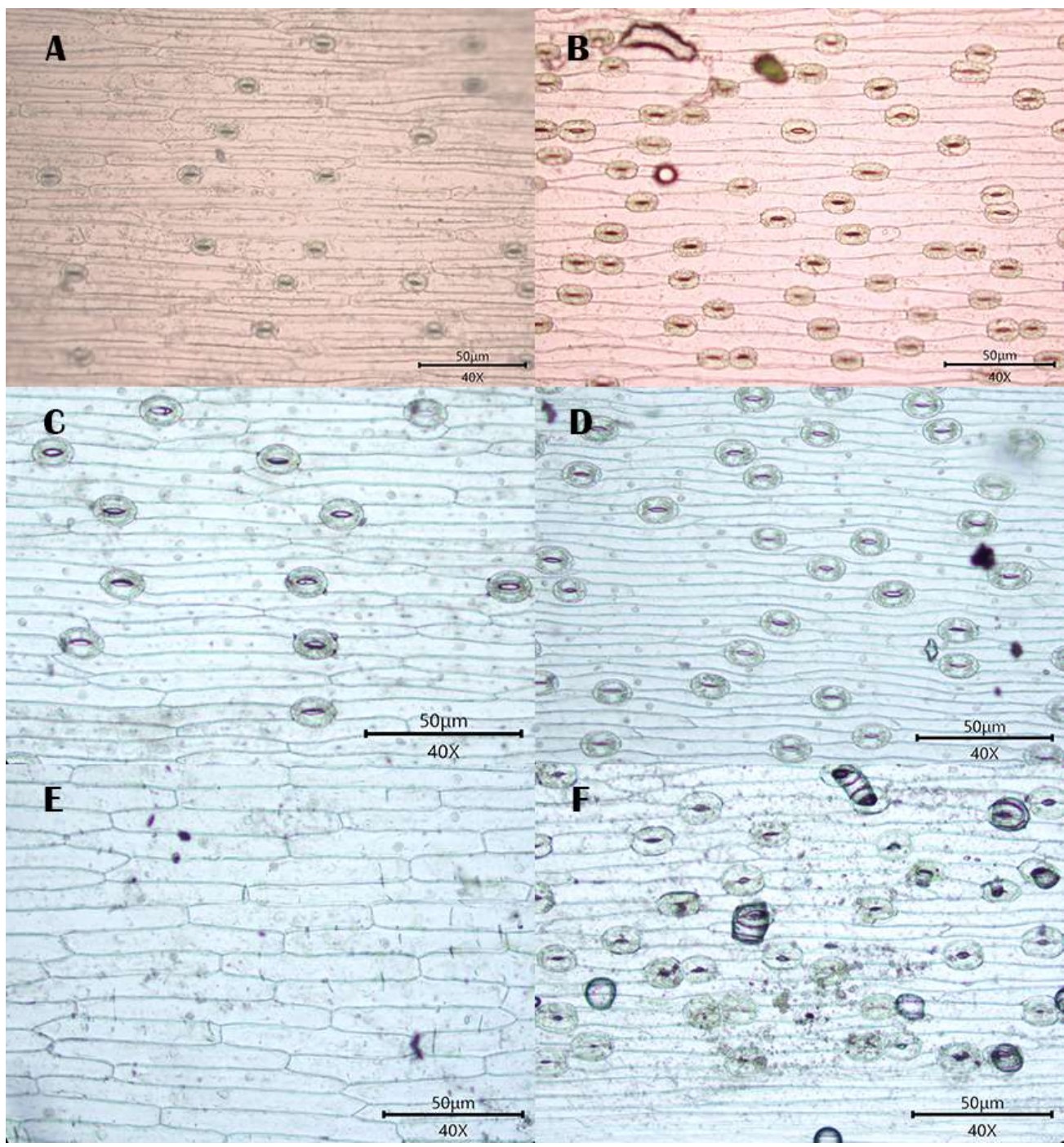

**Figure 6.** (**A**) Adaxial epidermis of *A. latifolia*; (**B**) abaxial epidermis of *A. latifolia*; (**C**) adaxial epidermis of *A. nanyueensis*; (**D**) abaxial epidermis of *A. nanyueensis*; (**E**) adaxial epidermis of *A. polymorpha*; (**F**) abaxial epidermis of *A. polymorpha*.

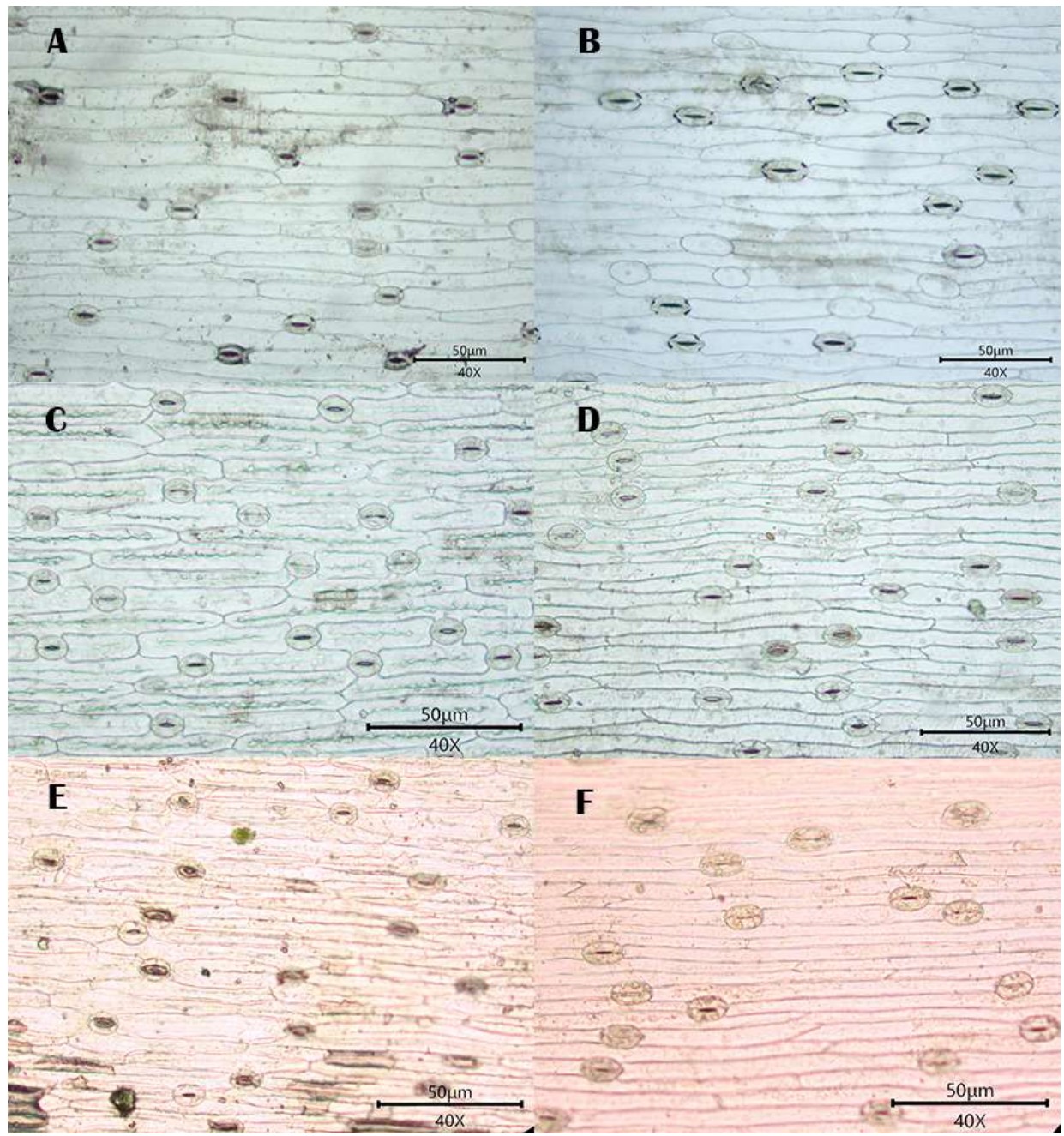

**Figure 7.** (**A**) Adaxial epidermis of *A. tianmuensis*; (**B**) abaxial epidermis of *A. tianmuensis*; (**C**) adaxial epidermis of *A. wanzhensis*; (**D**) abaxial epidermis of *A. wanzhensis*; (**E**) adaxial epidermis of *A. yunmengensis* ined.; (**F**) abaxial epidermis of *A. yunmengensis* ined.

**Table 2.** Leaf epidermal features (R: rectangle; L: long rectangle; N: near rectangle; D: rhombic; +++: dense stomata; +: sparse stomata; -: stomata absent; L: linear; P: paralinear; W: wavy).

| Species Name | Sample Number | Adaxial Epidermis | | | Abaxial Epidermis | | |
|---|---|---|---|---|---|---|---|
| | | Shape | Stoma | Anticlinal Wall | Shape | Stoma | Anticlinal Wall |
| *A. anhuiensis* (X.S.Shen) Christenh. | LJK61 | R, N | +++ | L | R | +++ | L |
| | LJK62 | R, N | +++ | L | R | +++ | L |
| | LJK63 | R, L | +++ | L | L | +++ | L |
| | WMZ1499 | R, L | +++ | L | L, D | +++ | L |

**Table 2.** *Cont.*

| Species Name | Sample Number | Adaxial Epidermis | | | Abaxial Epidermis | | |
|---|---|---|---|---|---|---|---|
| | | Shape | Stoma | Anticlinal Wall | Shape | Stoma | Anticlinal Wall |
| *A. baohuaensis* B.X.Han, Long Wang and G.Y.Lu, | LJK31 | R | +++ | L | L, R, N | +++ | L |
| | LP207885 | R | +++ | L | L, D | +++ | L |
| | LP207895 | R | +++ | L | R | +++ | L |
| | LP207904 | R | +++ | L | L | +++ | L |
| | WMZ1060 | R | +++ | L | R, D | +++ | P |
| | WMZ1417 | D | +++ | L | N | +++ | L |
| | WMZ1423 | L | +++ | L | L, D | +++ | L |
| *A. edulis* (Miq.) Honda | LJK12 | R | +++ | P | R, L | +++ | L |
| | LJK38 | N | +++ | P | N | +++ | L |
| | LJK54 | R, L | +++ | L | L | +++ | L |
| | LJK9 | L | +++ | P | L | +++ | L |
| | LP173055A | R | +++ | L | L | +++ | L |
| | LP173072 | R, L | +++ | P | L | +++ | L |
| | LP207881 | R, L | +++ | P | L | +++ | L |
| | LP207884 | R, L | +++ | L | L | +++ | L |
| | LP207887 | L | +++ | L | L, R, N | +++ | L |
| | LP207891 | R | +++ | P | L | +++ | P |
| | LP207905 | R | +++ | P | R, L | +++ | L |
| | LP207914 | L | +++ | L | L | +++ | L |
| | WMZ1488 | D | +++ | L | L | +++ | L |
| | WMZ1490 | R, L | +++ | L | L | +++ | L |
| *A. erythronioides* (Baker) D.Y.Tan and D.Y.Hong | LJK24 | N | + | W | L | +++ | L |
| | LJK26 | R | + | W | L | +++ | P |
| | LJK5 | R | +++ | W | L | +++ | L |
| *A. hejiaqingii* M.Z. Wang and P. Li, | LP173093 | L, N, D | + | L | R, L | +++ | L |
| | LP207883 | L | +++ | L | R | +++ | L |
| | WMZ1487 | L | +++ | P | L, D | +++ | L |
| | WMZ1489 | L | +++ | L | R | +++ | L |
| | WMZ1492 | L | +++ | L | R | +++ | L |
| | WMZ1714 | R, N | +++ | P | L | +++ | L |
| | WMZ1716 | R, N | +++ | P | L | +++ | L |
| | WMZ1721 | N | +++ | L | D, N | +++ | L |
| | ZXX19043 | R, N | +++ | P | L | +++ | L |
| *A. kuocangshanica* D.Y.Tan and D.Y.Hong | LJK22 | R | - | W | L | +++ | L |
| | LJK3 | R, N | - | W | R, L | +++ | L |
| | LJK7 | R, L | - | W | R, L | +++ | L |
| *A. kuocangshanica* × *A. latifolia* | WMZ1448 | R, L | +++ | L | L | +++ | L |
| *A. latifolia* (Makino) Honda | DSL01 | R, L | +++ | L | L | +++ | L |
| | WMZ1446 | R, L | +++ | L | L, D | +++ | L |
| *A. nanyueensis* P. Li and L.X. Liu | LP196220 | R | +++ | P | L | +++ | L |
| | WMZ1464 | R | +++ | P | L | +++ | L |
| *A. polymorpha* ined. | LJK10 | R | - | L | R, L | +++ | L |
| | LJK11 | L | + | L | L | +++ | L |
| | LJK21 | R | - | L | R, L | +++ | L |
| | LJK23 | R | - | P | R, D | +++ | L |
| | LP207908 | R | - | L | R, L | +++ | L |
| | WMZ1458 | L | + | L | L | +++ | L |
| *A. tianmuensis* P. Li and M.Z. Wang | LJK49 | R, L | +++ | L | D, R | +++ | L |
| | WMZ1473 | R | +++ | L | R, L | +++ | L |
| | WMZ1506 | R, L | +++ | L | D, R | +++ | L |
| *A. wanzhensis* Lu Q.Huang, B.X.Han and K.Zhang | LJK39 | L, D | +++ | L | L | +++ | L |
| | LJK41 | R | +++ | L | L | +++ | L |
| | LJK8 | D, R | +++ | L | L, D | +++ | L |
| | WMZ1455 | R | +++ | L | D, L | +++ | L |
| | WMZ1460 | R | +++ | L | D, L | +++ | L |
| *A. yunmengensis* ined. | WMZ1707-1L | R, N | +++ | L | L | +++ | L |
| | WMZ1707-2L | R, N | +++ | L | L | +++ | L |
| | WMZ1709-1L | R, L | +++ | L | L | +++ | L |
| | WMZ1709-3L | R, L | +++ | L | L | +++ | L |

## 3.2. Reconstruction of Epidermal Features

According to the evolutionary tree constructed based on the nuclear genes of *Amana* (unpublished data), the genus can be subdivided into three evolutionary branches: clade I ((*A. edulis, A. tianmunensis*), *A. nanyueensis*)), clade II ((*A. baohuaensis, A. wanzhensis*), ((*A. hejiaqingii, A. yunmengensis* ined.), *A. anhuiensis*))), and clade III (((*A. kuocangshanica, A. latifolia*), *A. erythronioides*), *A. polymorpha* ined.).

### 3.2.1. Reconstruction of Stomata Distribution

After reconstructing and analyzing the stomatal distribution traits of the adaxial epidermis, we found that the common ancestor of the three clades had dense stomata, and the ancestors of species in clade I and clade II were most likely to have had many stomata. In clade III, the common ancestor of *A. kuocangshanica* was considered to have dense stomata, the common ancestor of *A. polymorpha* ined. was most likely to have sparse stomata, and the common ancestor of *A. latifolia* and *A. erythronioides* had dense stomata. Through the observation of the *A. erythronioides* and *A. polymorpha* ined. evolutionary trees, we hypothesize that the general evolutionary trend for adaxial leaf epidermal stomatal distribution occurs from many to scarce to absent. In the abaxial epidermis, all samples had dense stomata and there is no need for reconstitution analysis (Figure 8).

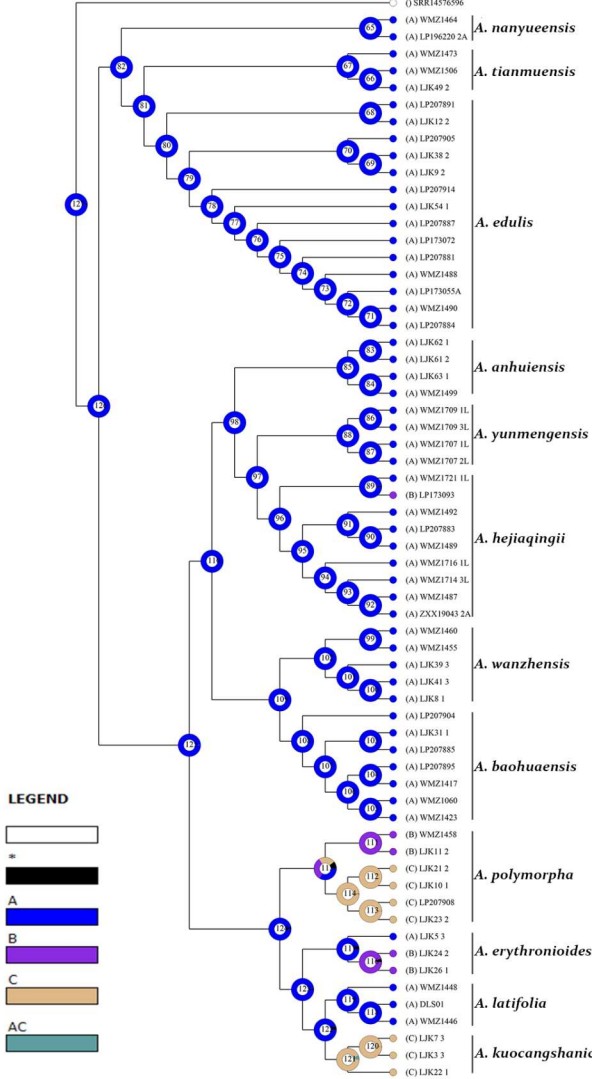

**Figure 8.** Reconstruction of adaxial epidermal stomatal distribution features (A: dense stomata; B: sparse stomata; C: stomata absent; white: no data; * (black): ambiguous).

### 3.2.2. Reconstruction of Epidermal Cell Morphology

After performing a morphological reconstruction analysis of adaxial epidermal cells, we found that the morphology of each sample was complex and variable. The common ancestor of *Amana* and each of the three clades were most likely rectangular. In the morphological analysis of abaxial epidermal cells, it was found that the common ancestor of this genus and each of the three clades were most likely to have long rectangular adaxial epidermal cells (Figures 9 and 10).

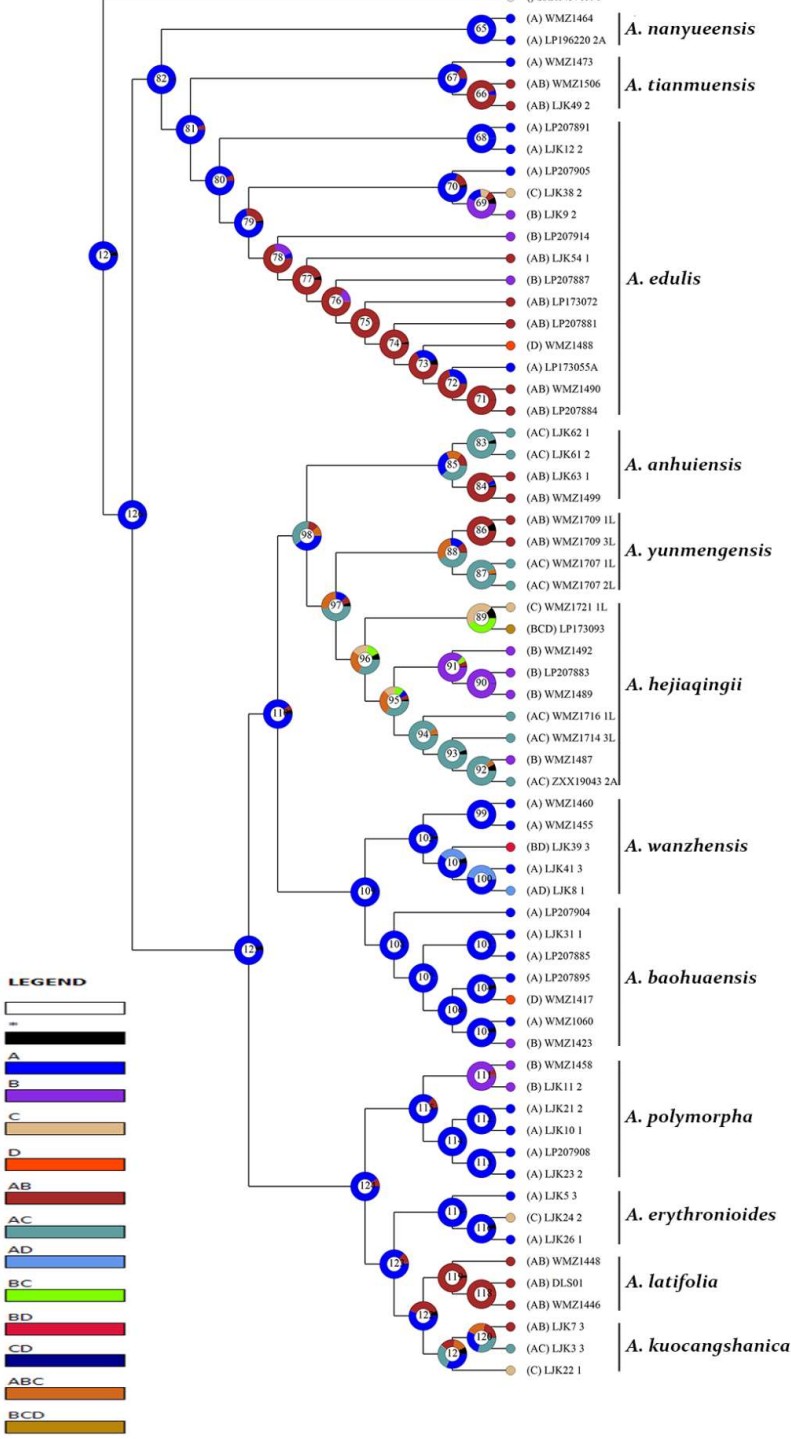

**Figure 9.** Reconstruction of cell morphological features of the adaxial epidermis (A: rectangle; B: long rectangle; C: near rectangle; D: rhombic; white: no data; * (black): ambiguous).

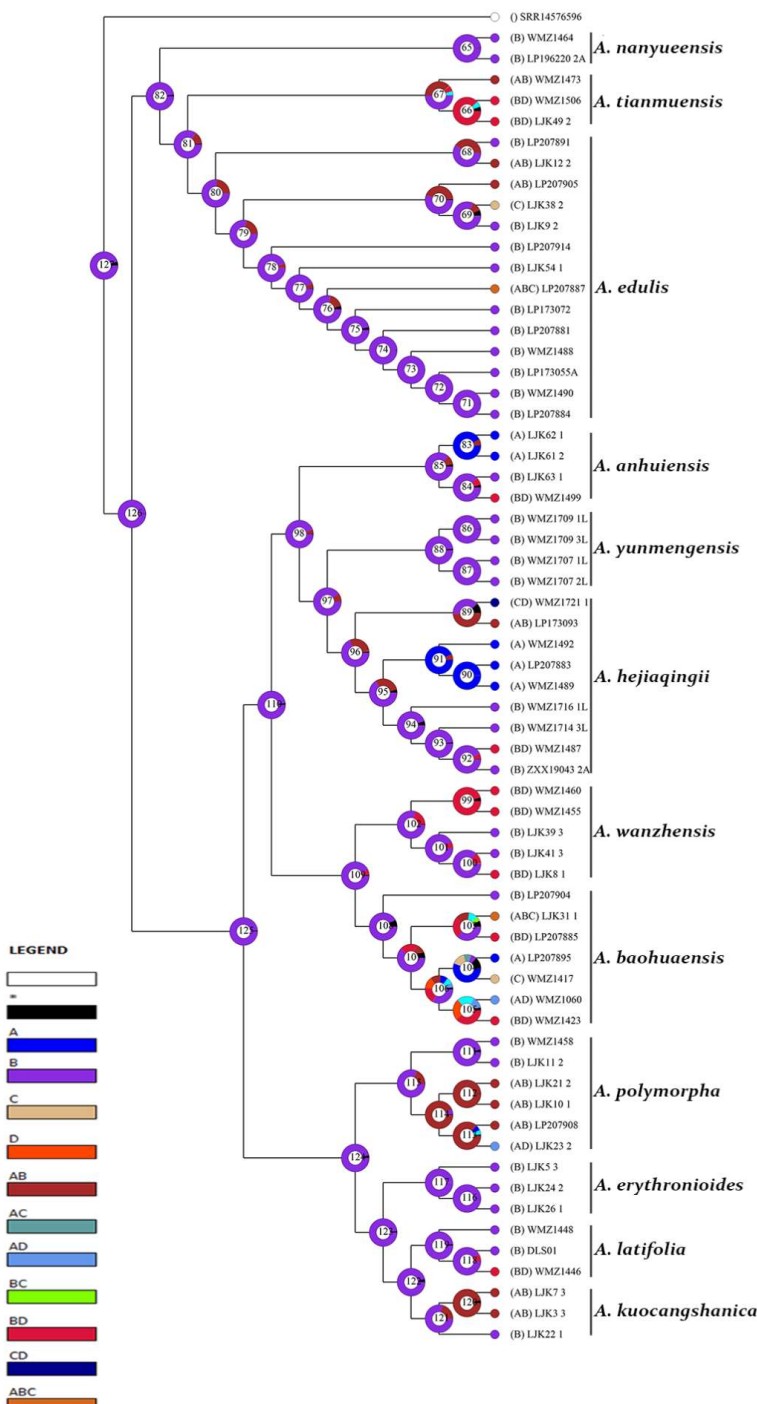

**Figure 10.** Reconstruction of cell morphological features of the abaxial epidermis (A: rectangle; B: long rectangle; C: near rectangle; D: rhombic; white: no data; * (black): ambiguous).

### 3.2.3. Reconstruction of the Anticlinal Wall Morphology of Epidermal Cells

After analyzing the morphological reconstruction of the anticlinal wall of the adaxial epidermal cells, it was found that the common ancestor of the three clades was most likely to have had a linear adaxial epidermal cell anticlinal wall, and the common ancestor of clade I was recovered as having a nearly rectilinear adaxial epidermal cell anticlinal wall. The most-recent common ancestors of *A. edulis* and *A. nanyueensis* were most likely to have had a nearly straight anticlinal wall, and the ancestor of *A. tianmuensis* was most likely to have had a rectilinear anticlinal wall. The ancestors of all species in clade II were most likely to have had rectilinear anticlinal walls. Except for one of the clades of *A. hejiaqingii*,

which shares the characteristics of a nearly linear anticlinal wall, all other species have a linear anticlinal wall. The common ancestor of clade III was most likely to have had a linear anticlinal wall, and the ancestor of *A. polymorpha* ined. was also most likely to have had a linear anticlinal wall. However, the situation is more complicated in another clade consisting of *A. erythronioides*, *A. latifolia*, and *A. kuocangshanica*. Since the ancestors of *A. kuocangshanica* and the *A. erythronioides* are most likely to have had a wavy anticlinal wall, and the ancestor of *A. latifolia* was most likely to have had a linear anticlinal wall, the character state for neither the common ancestor of *A. latifolia* and *A. kuocangshanica* nor that of the clade was clear, but it was most likely to have been wave-shaped. The anticlinal wall of abaxial epidermal ancestor cells was most likely straight for all species, and no reconstitution analysis was necessary (Figures 11 and 12).

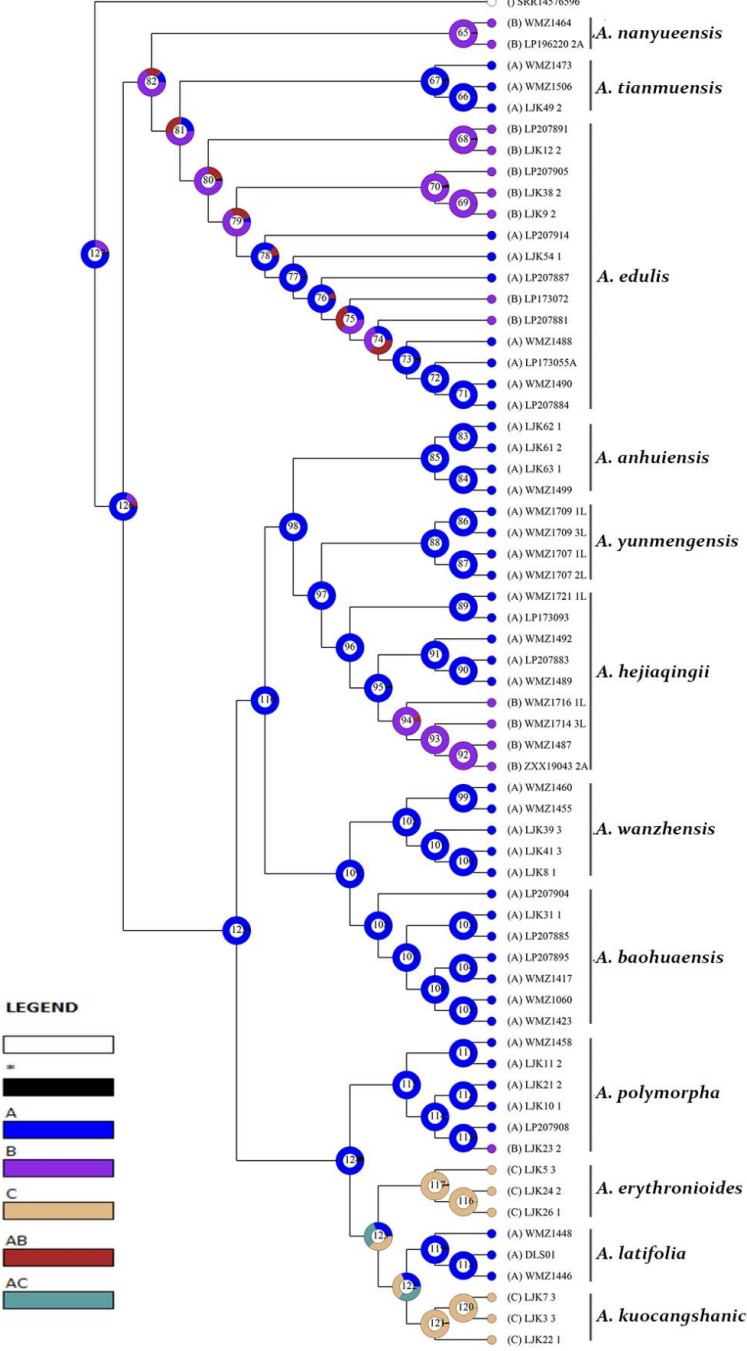

**Figure 11.** Reconstruction of features of the anticlinal wall of the adaxial epidermis (A: linear; B: paralinear; C: wavy; white: no data; * (black): ambiguous).

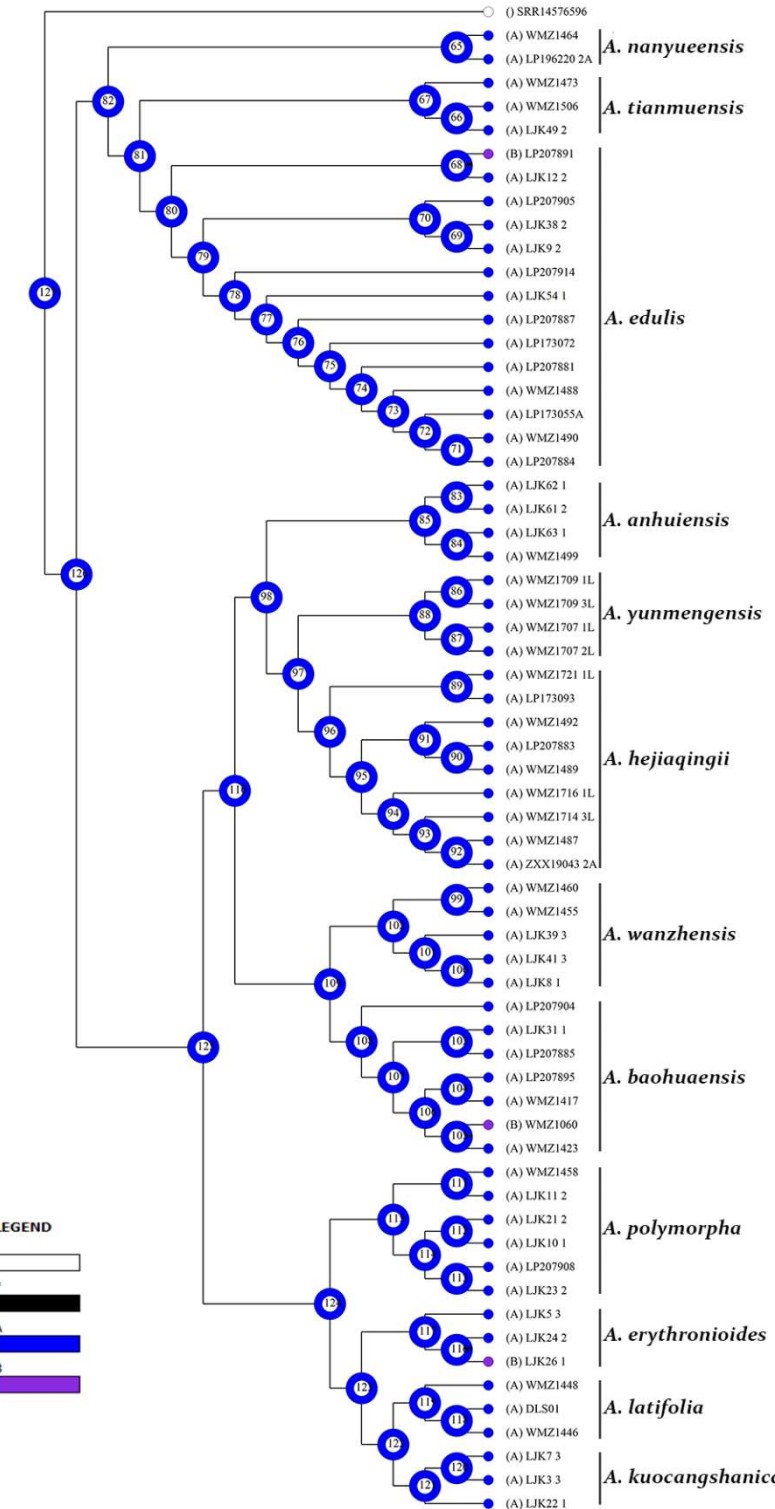

**Figure 12.** Reconstruction of features of the anticlinal wall of the abaxial epidermis (A: linear; B: paralinear; white: no data; * (black): ambiguous).

## 4. Discussion

### 4.1. The Significance of Leaf Epidermis Characteristics in the Classification Andrelationships of Amana

Microscopic observation of the leaf epidermis of the *Amana* was demonstrated to be useful for the identification of species classification and relationships. Consistent with previous studies, the adaxial epidermis of *A. kuocangshanica* lack stomata and the anticlinal

wall is wavy [5]. Additionally, our new data indicate that the *A. erythronioides* population collected by Tan et al. [12] from Mt. Huading, Tiantai County, Zhejiang Province should be identified as *A. polymorpha* ined. (Figures 8–12, unpublished data). Compared to Tan et al. [12], we found two stoma types present in *A. polymorpha*: sparse or absent. This difference was accomplished through the addition of more populations in our study. Therefore, only two species lack stomata on the adaxial surface (*A. polymorpha* ined. and *A. kuocangshanica*), and only two species display a wavy anticlinal wall (*A. erythronioides* and *A. kuocangshanica*). These species can be easily distinguished from other species based on these cellular characters, and confirm the relative relationship of these three species. Two unusual populations of *A. polymorpha* ined. (LJK11 and WMZ1458) from Banqiu Village in Zhuji City display special sparse stomata, possibly because they are hybrids of sympatric *A. wanzhensis* (unpublished data). At the same time, it is inferred that sample WMZ1448 (*A. kuocangshanica* × *A. latifolia*) is more similar to *A. latifolia*, rather than *A. kuocangshanica* (Figure 13), because the dense stomata in the adaxial epidermis and the linear anticlinal wall of the epidermal cells do not reflect the typical characteristics of *A. kuocangshanica*.

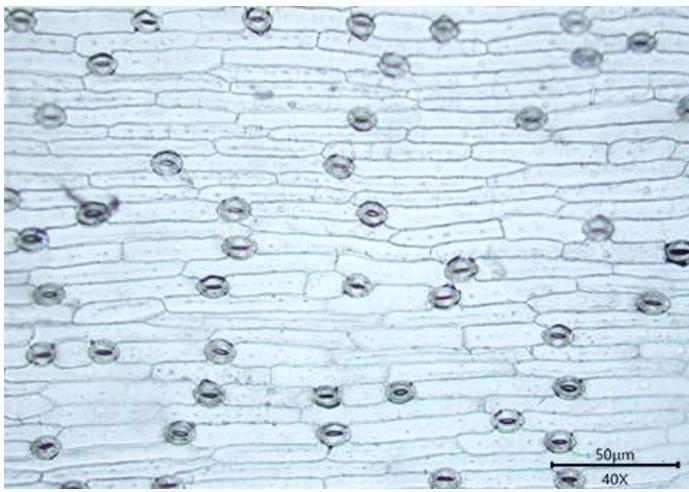

**Figure 13.** Adaxial epidermal micrograph of WMZ1448 (*A. kuocangshanica* × *A. latifolia*), which shows dense stomata and a linear anticlinal wall.

Stomata evolved during the transition of plants from aquatic to terrestrial, acting as the portals for water and gas exchange [17]. The growth environment of plants has a huge impact on the development of stomata [18]. For example, low light, temperature, and humidity, and high concentrations of carbon dioxide all contribute to decreased stomatal density [19–21]. Additionally, stomata development is also regulated by various hormones, such as brassinosteroid, abscisic acid, and auxin [22–24]. The mesophyll cells of most plants are differentiated into palisade mesophyll tissue and spongy mesophyll tissue. The palisade tissue is close to the adaxial epidermis and is the primarily location where photosynthesis occurs, and the spongy tissue is closer to the abaxial epidermis and mainly performs gas exchange; however, it can also perform photosynthesis [25]. To explain the mechanism behind the 'stomata absent' character state that we observed in *A. kuocangshanica* and *A. polymorpha*, more field investigations and experimental validation are needed.

### 4.2. Evolution of Leaf Epidermal Characteristics of Amana

According to the reconstruction of ancestral characters, the common ancestor of *Amana* is most likely to have had anticlinal vertical walls and dense stomata on the adaxial and abaxial epidermal cells. The morphology of the adaxial epidermal cells was most likely rectangular, while the abaxial epidermis cells were most likely long rectangular. The evolution of the abaxial epidermis is relatively conservative, with no significant changes in stomata and the morphology of the anticlinal wall of epidermal cells, which can be regarded as a common feature of *Amana*. The evolution rate of adaxial epidermis characteristics is

significantly faster, especially in the morphology of adaxial epidermis cells, which have many morphological changes and complex character combinations, and often differ within the same species; these characters can be used to study the evolutionary relationship between samples with very close genetic relationships. These shifts observed in these epidermal characteristics may be related to their living environment. *Amana* are spring ephemerals; this is a life strategy in which plants appear quickly in the early spring and complete their life cycle before canopy closure [26]. This rapid growth cycle requires high nutrient accumulation, which requires high photosynthetic efficiency. As is well known, $C_3$ plants have a lower ratio of the number of stomata in the adaxial and abaxial epidermis [27]. However, as a $C_3$ plant, most species of *Amana* possess stomata on both leaf sides, which may be related to the characteristics of rapid growth and strong photosynthetic capacity. It has been proven that the shift of stomata distribution from one leaf surface to both surfaces can improve gas exchange capacity and play an important role in regulating $CO_2$ diffusion via both stomata and mesophyll tissues, contributing to photosynthesis improvement [28–30]. Other studies have shown that smaller, denser leaf epidermal cells and stomata are more adapted to relatively arid and light-intensive habitats because they better regulate plant water metabolism [31]. The epidermal microscopic observations of this study provide a certain reference and basis for future morphological and physiological *Amana* studies.

## 5. Conclusions

In this study, we included 64 populations from all 12 *Amana* species and performed microscopic observations of their epidermal morphology. It was found that the morphology of the epidermal cells, perpendicular wall, and stomatal distribution were helpful for classification of *Amana* species and that the common ancestor of *Amana* is most likely to have had linear vertical walls and dense stomata on the adaxial and abaxial epidermal cells, which will deepen our understanding of the origin and evolution of the genus *Amana*.

**Author Contributions:** Conceptualization, P.L. (Pan Li); methodology, P.L. (Pan Li); investigation, M.W., M.C. and P.L. (Pan Li); experiment, X.Z., M.C. and P.L. (Pengcheng Luo); writing—original draft preparation, X.Z. and M.W.; writing—review and editing, M.C., P.L. (Pengcheng Luo), M.C.P. and P.L. (Pan Li); supervision, P.L. (Pan Li); resources, P.L. (Pan Li). All authors have read and agreed to the published version of the manuscript.

**Funding:** This research was funded by the National Natural Science Foundation of China (Grant No. 31970225), and the National Science and Technology Basic Project of China (Grant No. 2015FY110200).

**Data Availability Statement:** Not applicable.

**Acknowledgments:** We sincerely thank Yuantong Hou, Yonghua Zhang, Ruisen Lu, Yao Chen, Xuan Lu, Xinglv Xie, Huixia Cai, Jiaxian Dong, Zhenyu Jin, Xinxin Zhu, Fuhe Chen, Wenyun Wu, Difei Wu, Zhecheng Qi, Luxian Liu, Peizi He, Siyu Zhang, Jing Wu, Zhangshichang Zhu, Junke Li, Zongcai Liu, Xiaokai Fan, and Shenglu Zhang for helping with plant materials.

**Conflicts of Interest:** Author Pengcheng Luo was employed by the company Wuxi Biologics (Cayman) Inc. The remaining authors declare that the research was conducted in the absence of any commercial or financial relationships that could be construed as a potential conflict of interest.

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
