# Peer review of "Characteristics and Evolution of Leaf Epidermis in the Genus Amana Honda (Liliaceae)"

_2673-6500, doi:10.3390/taxonomy3030025_

Round 1

Reviewer 1 Report

Currently, a huge number of works on the molecular phylogeny of plants are published, but there is a lack of research on morphology and anatomy. This work contains a large amount of information on the structure of the leaf epidermis in the genus Amana, such data are necessary for studying taxonomy, evolution, and the relationship of plants with the environment. The article is well structured, the states of all characers are provided with illustrations, species descriptions are clearly stated. All the illustrationsin the article are of good quality. The authors carried out the reconstruction of characters using the molecular phylogenetic tree and put forward hypotheses about the evolution of the studied characters. The article deserves publication and will arouse the interest of readers.

Remarks.

1 In the abstract, the authors mention cryptic diversity in the genus Amana, but this issue is not covered in the main text of the article. We need to briefly talk about this in the introduction.

2 Material and Methods, a list of specimens, their geographical origin, and vouchers are required As in "Cytogeography of the East Asian Tulips (Amana, Liliaceae) 2022"

3 Describe the method for separating the leaf epidermis.

4 In fig. 1, 2, 3 you should write the names of the species photographed.

5, a short conclusion to the article is desirable.

6 The text contains misprints, in particular, in the section 3.1.1 (lines 73-81) the word "adaxial" is present three times and "abaxial" is missing.

Author Response

Authors Response

Point-by-point responses to the reviewers’ comments:

Reviewer 1

(1) In the abstract, the authors mention cryptic diversity in the genus Amana, but this issue is not covered in the main text of the article. We need to briefly talk about this in the introduction.

R: Thanks for your advice, we have added more information in the introduction.

(2) Material and Methods, a list of specimens, their geographical origin, and vouchers are required As in "Cytogeography of the East Asian Tulips (Amana, Liliaceae) 2022".

R: Thanks for your advice. We have added a table consisting the geographical information in revised MS.

(3) Describe the method for separating the leaf epidermis.

R: Thank you for your advice. We had added it in revised MS.

(4) In fig. 1, 2, 3 you should write the names of the species photographed.

R: Thank you for your advice. We had added the names in revised MS.

(5) a short conclusion to the article is desirable.

R: Thank you for your advice. We had added the conclusion in revised MS.

(6) The text contains misprints, in particular, in the section 3.1.1 (lines 73-81) the word "adaxial" is present three times and "abaxial" is missing.

R: Thank you for your advice. We had corrected it.

Reviewer 2 Report

The article presents data on epidermal structure as seen on paradermal sections in the genus Amana and the use of epidermal characters for systematics and taxonomy of the genus. The data themselves are interesting enough, but their presentation requires serious revision.

First of all, the abstract is very short and almost contains no information on the subject. The reader should receive, in brief, all the achievements without referring to the main text.

The authors accept 12 species of Amana—ten described and two undescribed in preparation. In the meantime, POWO accepts only six species https://powo.science.kew.org/taxon/urn:lsid:ipni.org:names:23968-1

This issue needs more careful explanation in the Introduction. Also, this important paper is not in the Literature Christenhusz, M.J.M., Govaerts, R., David, J.C., Hall, T., Borland, K., Roberts, P.S., Tuomisto, A., Buerki, S., Chase, M.W. & Fay, M.F. (2013). Tiptoe through the tulips - cultural history, molecular phylogenetics and classification of Tulipa (Liliaceae). Botanical Journal of the Linnean Society 172: 280-328.

The cell shapes are described as rectangular, long rectangular and nearly rectangular. The information on how to distinguish between these types using a ratio of dimensions should already appear in the Materials and Methods. In fig 2 I cannot distinguish between C and D. C is scored as nearly rectangular and D as rhombic. But to me, they look nearly the same, showing a similar situation to the rectangular and long rectangular types - central part of the cells is wider than the ends in both photos. The same is true for dense and sparse stomata. More than 10 and less than ten per view are OK, but what is the stomatal index (number of stomata per 1 mm2)? Also, the absolute dimensions, followed by appropriate statistical analyses are very important. There is no such information in the manuscript.

In Section 3.2 there is the statement that reconstructions are based on an unpublished nuclear gene tree topology. To trust this phylogeny, more information is needed about how it was inferred. Or use published data. What are the outgroups? There should be information on their epidermis as well.

Discussion lines 228-239. This is not surprising that stomata are absent in the adaxial epidermis of some species. This is typical for plants with bifacial leaves and a C3-photosynthetic pathway. More surprising, that other species possess stomata on both leaf sides.

Line 244 linear vertical walls – anticlinal?

Lines 258-260 ‘The dense stomata of the adaxial epidermis may benefit stronger heat resistance, conducive to an increase in transpiration strength, thereby effectively reducing the temperature of the leaf surface and protecting the leaf tissue from high temperature damage.’ – This is hard to accept. Many stomata on adaxial epidermis result in intensive transpiration (yes) but followed by intensive loss of water. Such plants are difficult to survive. Vast majority of the plants of hot habitats try to reduce transpiration. This issue needs more careful investigation and special analysis.

The article presents data on epidermal structure as seen on paradermal sections in the genus Amana and the use of epidermal characters for systematics and taxonomy of the genus. The data themselves are interesting enough, but their presentation requires serious revision.

First of all, the abstract is very short and almost contains no information on the subject. The reader should receive, in brief, all the achievements without referring to the main text.

The authors accept 12 species of Amana—ten described and two undescribed in preparation. In the meantime, POWO accepts only six species https://powo.science.kew.org/taxon/urn:lsid:ipni.org:names:23968-1

This issue needs more careful explanation in the Introduction. Also, this important paper is not in the Literature Christenhusz, M.J.M., Govaerts, R., David, J.C., Hall, T., Borland, K., Roberts, P.S., Tuomisto, A., Buerki, S., Chase, M.W. & Fay, M.F. (2013). Tiptoe through the tulips - cultural history, molecular phylogenetics and classification of Tulipa (Liliaceae). Botanical Journal of the Linnean Society 172: 280-328.

The cell shapes are described as rectangular, long rectangular and nearly rectangular. The information on how to distinguish between these types using a ratio of dimensions should already appear in the Materials and Methods. In fig 2 I cannot distinguish between C and D. C is scored as nearly rectangular and D as rhombic. But to me, they look nearly the same, showing a similar situation to the rectangular and long rectangular types - central part of the cells is wider than the ends in both photos. The same is true for dense and sparse stomata. More than 10 and less than ten per view are OK, but what is the stomatal index (number of stomata per 1 mm2)? Also, the absolute dimensions, followed by appropriate statistical analyses are very important. There is no such information in the manuscript.

In Section 3.2 there is the statement that reconstructions are based on an unpublished nuclear gene tree topology. To trust this phylogeny, more information is needed about how it was inferred. Or use published data. What are the outgroups? There should be information on their epidermis as well.

Discussion lines 228-239. This is not surprising that stomata are absent in the adaxial epidermis of some species. This is typical for plants with bifacial leaves and a C3-photosynthetic pathway. More surprising, that other species possess stomata on both leaf sides.

Line 244 linear vertical walls – anticlinal?

Lines 258-260 ‘The dense stomata of the adaxial epidermis may benefit stronger heat resistance, conducive to an increase in transpiration strength, thereby effectively reducing the temperature of the leaf surface and protecting the leaf tissue from high temperature damage.’ – This is hard to accept. Many stomata on adaxial epidermis result in intensive transpiration (yes) but followed by intensive loss of water. Such plants are difficult to survive. Vast majority of the plants of hot habitats try to reduce transpiration. This issue needs more careful investigation and special analysis.

Author Response

(1) First of all, the abstract is very short and almost contains no information on the subject. The reader should receive, in brief, all the achievements without referring to the main text.

R: Thanks for your advice, we have added more information in the abstract.

(2) The authors accept 12 species of Amana—ten described and two undescribed in preparation. In the meantime, POWO accepts only six species https://powo.science.kew.org/taxon/urn:lsid:ipni.org:names:23968-1

R: The genus Amana comprises about 12 perennial herbaceous species, of which seven has already been published: A. anhuiensis (X.S. Shen) Christenh., A. baohuaensis B.X. Han, Long Wang & G.Y. Lu, A. edulis (Miq.) Honda, A. erythronioides (Baker) D.Y. Tan & D.Y. Hong, A. kuocangshanica D.Y. Tan & D.Y. Hong, A. wanzhensis Lu Q. Huang, B.X. Han & K. Zhang and A. latifolia (Makino) Honda, and three newly published: A. nanyueensis P. Li & L.X. Liu, A. tianmuensis P. Li & M.Z. Wang, and A. hejiaqingii M.Z. Wang & P. Li, and two new species in review.: A. polymorpha ined. and A. yunmengensis ined.). POWO treats A. latifolia as a synonym of A. erythronioides, but both morphological characteristics and transcriptomic phylogeny support they are two distinct species in our unpublished research.

(3) This issue needs more careful explanation in the Introduction. Also, this important paper is not in the Literature Christenhusz, M.J.M., Govaerts, R., David, J.C., Hall, T., Borland, K., Roberts, P.S., Tuomisto, A., Buerki, S., Chase, M.W. & Fay, M.F. (2013). Tiptoe through the tulips - cultural history, molecular phylogenetics and classification of Tulipa (Liliaceae). Botanical Journal of the Linnean Society 172: 280-328.

R: Thanks for your advice, we have added it in the reference.

(4) The cell shapes are described as rectangular, long rectangular and nearly rectangular. The information on how to distinguish between these types using a ratio of dimensions should already appear in the Materials and Methods. In fig 2 I cannot distinguish between C and D. C is scored as nearly rectangular and D as rhombic. But to me, they look nearly the same, showing a similar situation to the rectangular and long rectangular types - central part of the cells is wider than the ends in both photos. The same is true for dense and sparse stomata. More than 10 and less than ten per view are OK, but what is the stomatal index (number of stomata per 1 mm2)? Also, the absolute dimensions, followed by appropriate statistical analyses are very important. There is no such information in the manuscript.

R: As to the rhombus and near-rectangular, what we think of as rhombus is the cells whose anticlinal wall is straight and the shape is close to a regular rhombus, while near-rectangular can summarize all cells with irregular edges, overall distortion or difficult to describe (when such cells appeared in large numbers, we had to use a word to describe it). And we did not count the stomatal index, because at the beginning of the experiment, we first noticed the lack of stomata in the entire upper epidermis rather than density, so the recorded information is the presence or absence of stomatal distribution. However, with the deepening of the research, we found some samples with stomatal distribution but the stomatal density was significantly lower than other records, so the trait of "sparse stomatal distribution" was sorted out separately.

(5) In Section 3.2 there is the statement that reconstructions are based on an unpublished nuclear gene tree topology. To trust this phylogeny, more information is needed about how it was inferred. Or use published data. What are the outgroups? There should be information on their epidermis as well.

R: Thanks for your advice, we have added more details in the revised MS.

(6) Discussion lines 228-239. This is not surprising that stomata are absent in the adaxial epidermis of some species. This is typical for plants with bifacial leaves and a C3-photosynthetic pathway. More surprising, that other species possess stomata on both leaf sides.

R: Thanks for your advice. After reviewing the literature, we guessed that this characteristic should be related to the characteristics of the early spring ephemeral plants of Amana. That is, the photosynthetic rate can be increased through high-efficiency gas exchange, so that the growth cycle can be completed quickly.

(7) Line 244 linear vertical walls – anticlinal?

R: Thanks for your advice, we have made revision here.

(8) Lines 258-260 ‘The dense stomata of the adaxial epidermis may benefit stronger heat resistance, conducive to an increase in transpiration strength, thereby effectively reducing the temperature of the leaf surface and protecting the leaf tissue from high temperature damage.’ – This is hard to accept. Many stomata on adaxial epidermis result in intensive transpiration (yes) but followed by intensive loss of water. Such plants are difficult to survive. Vast majority of the plants of hot habitats try to reduce transpiration. This issue needs more careful investigation and special analysis.

R: Thanks for your advice. We have removed this description and added a conjecture about stomata on both leaf sides can increase photosynthetic efficiency in revised MS.

Reviewer 3 Report

Missing Voucher Specimens,

Where they are stored

Check grammar and professional erminology

Author Response

(1) Missing Voucher Specimens. Where they are stored

R: We have added this information in revised MS.

(2) Comments on the Quality of English Language. Check grammar and professional terminology.

R: Thanks for your advice. We have made revision.

Reviewer 4 Report

Authors have included 64 populations from all 12 Amana species and performed microscopic observations of their epidermal morphology in the manuscript entitled, “Characteristics and evolution of leaf epidermis in the genus Amana Honda (Liliaceae)”. Minor corrections are required;

Distinctive morphoanatomical features that led to the segregation of Amana from Tulipa should be included in the introduction.

Line 26: Fan et al., 2014 used epidermis…. Please follow reference format of the journal  i.e, Fan et al. [1] used epidermis

This will lead to revision of all reference numbers in the manuscript.

Line 38: Amana edulis author citation?

Line 39: havestomata…. have stomata

Line 40: A. kuocangshanica a.cit?

Line 44: 12Amanaspecies…spacing issue

Line 76: a. citations missing in text…A. erythronioides , A. edulis, A. tianmunensis, A. nanyueensis, A. baohuaensis, A. wanzhensis, A. hejiaqingii,  A. anhuiensis, A. latifolia

Line 213: Tan et al. (2007)…. Please follow reference format of the journal  

Line 215: Tan et al. (2007)…. Please follow reference format of the journal

Alignment of figures should be according to the journal format.

Author Response

(1) Distinctive morphoanatomical features that led to the segregation of Amana from Tulipa should be included in the introduction.

R: Thanks for your advice. We have added this information in revised MS.

(2) Line 26: Fan et al., 2014 used epidermis…. Please follow reference format of the journal  i.e, Fan et al. [1] used epidermis.

R: Thanks for your advice. We have corrected it.

(3) Line 38: Amana edulis author citation?

R: Thanks for your advice. We have added it.

(4) Line 39: havestomata…. have stomata.

R: Thanks for your advice. We have corrected it.

(5) Line 40: A. kuocangshanica a.cit?

R: Thanks for your advice. We have added it.

(6) Line 44: 12Amanaspecies…spacing issue

R: Thanks for your advice. We have corrected it.

(7) Line 76: a. citations missing in text…A. erythronioides , A. edulis, A. tianmunensis, A. nanyueensis, A. baohuaensis, A. wanzhensis, A. hejiaqingii,  A. anhuiensis, A. latifolia.

R: Thanks for your advice. We have added it.

(8) Line 213: Tan et al. (2007)…. Please follow reference format of the journal

Line 215: Tan et al. (2007)…. Please follow reference format of the journal

R: Thanks for your advice. We have corrected it.

Round 2

Reviewer 2 Report

The revised manuscript looks much better after the revision. The authors responded to all my points and comments. The paper can be accepted in its current form.

Author Response

Thank you for your valuable advice to make this article better!